# Enhancing Early Lung Cancer Diagnosis: Predicting Lung Nodule Progression in Follow-Up Low-Dose CT Scan with Deep Generative Model

**DOI:** 10.3390/cancers16122229

**Published:** 2024-06-15

**Authors:** Yifan Wang, Chuan Zhou, Lei Ying, Heang-Ping Chan, Elizabeth Lee, Aamer Chughtai, Lubomir M. Hadjiiski, Ella A. Kazerooni

**Affiliations:** 1Department of Radiology, The University of Michigan Medical School, Ann Arbor, MI 48109-0904, USA; wangyfan@umich.edu (Y.W.); chanhp@umich.edu (H.-P.C.); echaripa@umich.edu (E.L.); chughta@ccf.org (A.C.); lhadjisk@med.umich.edu (L.M.H.); ellakaz@umich.edu (E.A.K.); 2Department of Electrical Engineering and Computer Science, The University of Michigan, Ann Arbor, MI 48109-2122, USA; leiying@umich.edu; 3Diagnostic Radiology, Cleveland Clinic, Cleveland, OH 44195, USA; 4Department of Internal Medicine, The University of Michigan Medical School, Ann Arbor, MI 48109-0904, USA

**Keywords:** lung cancer, early diagnosis, generative AI, nodule growth prediction, deep learning

## Abstract

**Simple Summary:**

Detecting lung cancer early and initiating treatment promptly can greatly enhance patient outcomes. While low-dose computed tomography (LDCT) screening aids in identifying lung cancer at an early stage, there is a risk of diagnostic delays as patients await follow-up scans. To mitigate this challenge, we developed a deep predictive model leveraging generative AI methods to forecast nodule growth patterns in follow-up LDCT scans based on baseline LDCT scans. Our findings illustrated that utilizing the predicted follow-up nodule images generated by our model during baseline screening improved diagnostic accuracy compared to using baseline nodules alone and achieved comparable performance with using real follow-up nodules. This demonstrated the potential of employing deep generative models to forecast nodule appearance in follow-up imaging from baseline LDCT scans, thereby enhancing risk assessment during initial screening.

**Abstract:**

Early diagnosis of lung cancer can significantly improve patient outcomes. We developed a Growth Predictive model based on the Wasserstein Generative Adversarial Network framework (GP-WGAN) to predict the nodule growth patterns in the follow-up LDCT scans. The GP-WGAN was trained with a training set (N = 776) containing 1121 pairs of nodule images with about 1-year intervals and deployed to an independent test set of 450 nodules on baseline LDCT scans to predict nodule images (GP-nodules) in their 1-year follow-up scans. The 450 GP-nodules were finally classified as malignant or benign by a lung cancer risk prediction (LCRP) model, achieving a test AUC of 0.827 ± 0.028, which was comparable to the AUC of 0.862 ± 0.028 achieved by the same LCRP model classifying real follow-up nodule images (*p* = 0.071). The net reclassification index yielded consistent outcomes (NRI = 0.04; *p* = 0.62). Other baseline methods, including Lung-RADS and the Brock model, achieved significantly lower performance (*p* < 0.05). The results demonstrated that the GP-nodules predicted by our GP-WGAN model achieved comparable performance with the nodules in the real follow-up scans for lung cancer diagnosis, indicating the potential to detect lung cancer earlier when coupled with accelerated clinical management versus the current approach of waiting until the next screening exam.

## 1. Introduction

Lung cancer is a leading cause of cancer-related death for men and women worldwide, with a poor overall 5-year survival rate of 22.9% [1]. Early diagnosis and treatment of lung cancer can significantly improve patient survival. For patients diagnosed at an early stage, the five-year survival rate can reach about 61.2% [2]. In contrast, curative treatments are no longer effective for advanced-stage lung cancer, resulting in a five-year survival rate of only 16.0%.

Studies have shown that screening with low-dose computed tomography (LDCT) can help detect lung cancer at an early stage and reduce lung cancer mortality in people at high risk [3]. However, the LDCT screening also detects many nodules that are considered to be indeterminate. Although the nodule size, shape, and density manifested on the LDCT images are correlated with the risk of malignancy, a single LDCT scan at initial screening (baseline) may not provide definitive information, so follow-up is often required to evaluate the stability of the nodule over time [4]. With follow-up LDCT scans, radiologists can visualize how the nodule changes over time and make more accurate and informed decisions regarding patient management, including the need for further evaluation, biopsies, or follow-up scans.

Besides the increased cost and additional radiation, patients may experience delays in diagnosis as they wait for follow-up LDCT scans. A potential method is to harness the power of advanced artificial intelligence (AI) techniques to analyze the lung nodule characteristics manifested on CT scans and predict their progression in subsequent follow-up CT scans. However, the prediction of lung nodule evolution is challenging because of the heterogeneous and multifactorial nature of lung nodules [5]. Traditional mathematical models have limited predictive power as they usually use simplistic formulations, for example, linear, quadratic, power-law, and exponential models [6,7], and only take into consideration physical factors of nodules such as size, mass, and volume, and thus may not be able to adequately capture the biological complexity of nodules. Machine learning-based radiomics methods have demonstrated the potential to predict the likelihood of lung nodule growth [8,9]. However, these methods relied on effective radiomic features [8,9], including manually engineered features [10,11], that demand expertise in both engineering and domain knowledge to interpret data patterns and design feature extractors [12]. Leveraging the advances of AI techniques, deep-learning-based methods such as Convolutional Long Short-Term Memory (Conv-LSTM) [13], U-Nets [14], and deep spatial transformation process [15] demonstrated the ability to characterize complex image patterns of nodules detected in CT scans. While the aforementioned methods have shown promise in predicting the risk of malignancy and the appearance of nodules in future exams for lung cancer diagnosis, they required prior CT scans to detect interval changes in the nodule between current and prior CT scans. This prerequisite renders these methods useless when prior scans are unavailable, and indeterminate nodules have to undergo at least one follow-up scan.

Generative models have been increasingly explored in medical imaging applications for different tasks, such as image synthesis, reconstruction, noise reduction, segmentation, and classification [16,17,18,19,20]. The generative adversarial network (GAN) has emerged as a powerful tool for data augmentation by generating new samples that are similar to the original data, for example, increasing the size and diversity of a training dataset for classification and image segmentation [18,19]. It has also been employed for cross-modality image synthesis that can transform the image of one modality to another modality, for example, by synthesizing magnetic resonance images from CT images [21].

In this study, we developed a deep-learning-based generative model to discover patterns of nodule growth on longitudinal CT scans during the training process and predict their progression without requiring prior scans at deployment. With time serial data collected from the National Lung Screening Trial (NLST) study [3], we developed a Growth Predictive model based on the Wasserstein Generative Adversarial Network training framework, referred to as the GP-WGAN model, to learn the growth pattern from pairs of nodule images in LDCT scans acquired at baseline (T0) and follow-up screening years (T1, T2). We designed new loss functions to guide detailed pixel-wise synthesis, preserve structural similarity, and enhance the perceptual quality of the generated nodules with reference to the follow-up images in future LDCT scans.

To the best of our knowledge, we are the first to develop a deep generative model with the ability to predict lung nodule images in future follow-up LDCT scans from images of current-year LDCT scans. Through the combination of four different loss functions that were designed to measure specific errors, the follow-up nodule images predicted from baseline scans by the GP-WGAN model achieved comparable performance with the actual nodules in follow-up LDCT scans in predicting lung cancer risks. This demonstrates the feasibility of using deep generative models to improve the clinical management of screening-detected pulmonary nodules, facilitating earlier diagnosis of lung cancer.

## 2. Materials and Methods

### 2.1. Data Sets

This retrospective study was approved by the Institutional Review Board (IRB), and informed consent was waived. The NLST was a randomized controlled trial that randomly assigned participants to receive three annual screenings (T0, T1, and T2) with either LDCT or chest X-rays. The participants who were diagnosed with lung cancer through biopsy confirmation and required treatment were not offered further screening LDCT scans. With permission from the NLST, we collected 2500 anonymized subjects containing digital LDCT files, including all 639 NLST-reported biopsy-confirmed lung cancer cases, and 1861 randomly selected benign nodule cases based on 3 years annual LDCT exams and/or up to seven years of non-imaging follow-up. More information about the LDCT scans is described in Appendix A. Among the 2500 subjects, 1226 subjects who had at least one nodule with a size ranging from 4 mm to 30 mm found in their first-year baseline LDCT scans and had corresponding follow-up LDCT scans were included in this study. Of these 1226 subjects, 218 were diagnosed with lung cancer and 1008 were negative. We randomly split 1226 subjects into a training/validation set comprising 776 subjects (165 positive and 611 negative) and a test set containing 450 subjects (53 positive and 397 negative) (Table 1). From the 776 subjects in the training/validation set who underwent baseline and annual follow-up scans up to 2 years, a total of 1121 pairs of image patches (776 pairs between T0 and T1, 345 pairs between T1 and T2) containing nodules (223 positive and 898 negative) from LDCT scans were used to develop the GP-WGAN model. In the test set, only the baseline scans and their corresponding T1 scans were included, comprising a total of 450 (53 positive and 397 negative) pairs of nodule image patches. Among the 53 positive subjects, 42 and 11 were diagnosed with lung cancer at T1 and T2 follow-up LDCT screening, respectively. All 397 negative subjects had undergone 2 years of follow-up LDCT exams.

### 2.2. Study Subject Characteristics

The distribution of the NLST documented subject demographic data and clinical radiologic factors was summarized in Table 1.

### 2.3. Data Preparation

An experienced cardiothoracic radiologist re-examined each NLST-documented lung nodule and manually marked the corresponding nodule centers on the baseline and follow-up scans for each subject. In a subject with multiple nodules detected at the baseline scan, the nodule with the largest size/growth during follow-up was identified by the radiologist and used in this study [8,9]. All LDCT scans were resampled to isotropic volumes with a voxel size of 0.5 × 0.5 × 0.5 mm^3^ using the 3D spline interpolation method [22]. An experienced cardiothoracic radiologist manually selected the slice for each nodule to ensure that the selected slice correctly manifested the nodule’s characteristics. A 2D region of interest (ROI) on the slide with a side length of 32 mm centered at the radiologist’s manually marked nodule center was extracted and used as the nodule image patch.

### 2.4. Growth Predictive Model Based on the Wasserstein Generative Adversarial Network

We developed a GP-WGAN model with a deep predictor network and a deep discriminator network to predict a nodule image in the follow-up LDCT scan from their baseline LDCT scan. Figure 1 shows the adversarial framework for the training of the GP-WGAN model. The predictor network was implemented with a modified U-Net architecture [23], and the discriminator network was constructed with a five-layer deep convolutional neural network (DCNN) structure. The structures of the predictor network and the discriminator network are described in Appendix B.

The pairs of nodule ROI images and their corresponding follow-up images at 1-year intervals (T0–T1 pairs or T1–T2 pairs if available) were input to the predictor and discriminator networks to train the GP-WGAN model. The real follow-up images were used as the target to guide the predictor in generating images that “mimic” the nodules in their follow-up scans. With the goal of minimizing the Wasserstein adversarial loss, the task of the discriminator was to distinguish the real follow-up images from the predicted images, provide feedback (losses) to the predictor network, and update the weights of the networks. We trained the predictor and discriminator networks iteratively by minimizing a weighted combination of losses. The predictor aimed to synthesize more realistic images that the discriminator could not distinguish from the real follow-up images. The discriminator, on the other hand, continuously improved its ability to distinguish between the real and predicted images. This adversarial training process created a feedback loop between the predictor and discriminator. Once the GP-WGAN model was trained, only the predictor network with frozen weights was used to predict the nodule images in future LDCT scans. The discriminator network was no longer needed during deployment.

### 2.5. Generative Loss Function

We designed a generative loss function ($L_{G}$) in our GP-WGAN model that leveraged $L_{1}$ loss for pixel-wise synthesis accuracy, structural similarity index loss ($L_{SSIM}$) [24] for preserving structure information, a learned perceptual [25] loss ($L_{LP}$) for capturing high-level visual quality, and an adversarial loss ($L_{A}$) to ensure realistic outputs:(1)$$L_{G}=\lambda_{1}L_{1}+\lambda_{2}L_{SSIM}+\lambda_{3}L_{LP}+\lambda_{4}L_{A},$$
where $\lambda_{1}$, $\lambda_{2}$, $\lambda_{3}$, and $\lambda_{4}$ were hyper-parameters that weighted the four losses. The calculation of $L_{G}$ involved using the pair of the real nodule image *X* and its follow-up $X_{F}$, and the “mimic” image $G\left(X\right)$ generated by the predictor network. More detailed descriptions of each loss are included in Appendix C.

### 2.6. Discriminator Loss Function

The loss of the discriminator was measured by the Wasserstein distance with gradient penalty [26] as follows:(2)$$L_{D}=-E_{G\left(X\right)}\left[D\left(G\left(X\right)\right)\right]+E_{X_{F}}\left[D\left(X_{F}\right)\right]+\lambda_{D}E_{\hat{x}}\left[{\left(\left|\left|\nabla_{\hat{x}}D\left(\hat{x}\right)\right|\right|-1\right)}^{2}\right],$$
where $\hat{x}=tX_{F}+\left(1-t\right)G\left(X\right)$ was an interpolated image, $t∼Unif\left(\left[0,1\right]\right)$, $\lambda_{D}$ was the penalty weight, and $\nabla_{\hat{x}}$ represented the gradient with respect to $\hat{x}$.

### 2.7. Performance Evaluation and Statistical Analysis

Using the independent test set consisting of 450 subjects (53 positive and 397 negative), we evaluated the potential of the GP-WGAN model in improving the early diagnosis of lung cancer by predicting follow-up nodule images from baseline LDCT nodule images. We deployed the trained GP-WGAN to the ROI images of the 450 nodules in the baseline LDCT scans to predict the GP-nodules in the 1-year follow-up LDCT scans. A lung cancer risk prediction (LCRP) model developed with serial radiomics-based reinforcement learning [27] was directly applied to the 450 GP-nodules without retraining to predict the risk of malignancy for each nodule. We compared the LCRP model to predict the risk for “virtual” GP-nodules versus the real nodules from the baseline and 1-year follow-up LDCT scans. These predicted risks were also compared to those predicted by the Lung-RADS and the Brock model estimated from the patient’s risk factors [28,29], such as demographic data and radiologic descriptions of nodules provided by the NLST dataset [3].

The receiver operating characteristic (ROC) analysis [30,31,32] and reclassification benefit analysis [33] were used for performance evaluation. The ROC curves of the LCRP model and the Brock model for nodule classification were compared using the method of DeLong et al. [34]. The Hochberg correction [35] was employed to adjust the *p*-values for multiple comparisons. The *p*-values were adjusted using the R software function “p.adjust”, and less than 0.05 after adjustment was considered statistically significant. The net reclassification index (NRI) [34] was used to assess the net gain or loss from the correct and incorrect risk escalation or de-escalation by the proposed method (i.e., LCRP model with GP-nodules as input) in comparison with the reference models that used the baseline nodules to stratify cancer risk. More detailed information and limitations of NRI are provided in Appendix D. The groups of low-, medium-, and high-risk were stratified by thresholding the Lung-RADS scores (<3, =3, and >3) derived from previous studies [36,37] and the scores of the Brock model (<0.0117, [0.0117, 0.10], >0.10) derived from the British Thoracic Society guideline [38,39]. The thresholds of the LCRP models (<0.45, [0.45, 0.81], >0.81) were determined so that the subgroup sizes by the LCRP model classifying GP-nodules would align with the subgroup sizes stratified by the Brock model [33]. The statistical significance of NRI was tested by Z-statistic [40]. The GP-WGAN model was developed using Python 3.6.9. and PyTorch 1.8.1. The ROC and other statistical analyses were performed by using the statistical software package ORDBM MRMC 3.0 in Java [41].

## 3. Results

Figure 2 shows examples of six lung nodules on the baseline (T0), 1-year follow-up (T1) LDCT scans, and GP-WGAN predicted GP-nodules.

Compared with the follow-up nodule images in the independent test set, GP-nodules achieved a Mean Square Error (MSE) of 0.024 and a Structural Similarity Index (SSIM) of 0.860. Figure 3 shows the test ROC curves, and Table 2 shows the test AUC achieved by our LCRP model and Brock model for the classification of 450 nodules in the independent test set. The LCRP model classifying real follow-up (T1) nodules achieved a test AUC of 0.862 ± 0.028. In comparison, the LCRP model classifying GP-nodules that were predicted from the baseline nodules achieved a comparable test AUC of 0.827 ± 0.028 (*p* = 0.071). The test AUC by both the LCRP model with baseline nodules (0.805 ± 0.031; *p* = 0.024) and the Brock model with baseline nodules (0.754 ± 0.035; *p* < 0.001) were significantly lower than the LCRP model with real follow-up nodules.

For early diagnosis of lung nodules from baseline scans, the LCRP model classifying GP-nodules achieved a significantly higher performance (*p* = 0.043) compared to the Brock model using real baseline nodules and only achieved a comparable performance (*p* = 0.099) compared to the LCRP model classifying real baseline nodules.

Table 2 shows that, for the classification of three subgroups of nodules with predominantly solid attenuation, spiculated margin, or nodules with size ranging from 6 to 14 mm in diameter, the LCRP model with real follow-up nodules achieved a test AUC of 0.864 ± 0.034, 0.922 ± 0.037, and 0.826 ± 0.039, respectively. In comparison, the LCRP model with GP-nodules achieved a comparable test AUC of 0.828 ± 0.037 (*p* = 0.091), 0.850 ± 0.055 (*p* = 0.150), and 0.782 ± 0.041 (*p =* 0.077), respectively. The test AUCs achieved by both the LCRP model with baseline nodules and the Brock model with baseline nodules were all significantly lower than the LCRP model with real follow-up nodules.

Table 2 also shows that, for early diagnosis of the same three subgroups, the LCRP model with the predicted GP-nodules achieved a significantly higher performance (*p =* 0.045, *p* < 0.001, *p =* 0.048, respectively) than the Brock model with real baseline nodules, whereas the LCRP model with real baseline nodules was comparable to the Brock model for solid nodules (*p =* 0.249) and nodules with size from 6 to 14 mm in diameter (*p =* 0.201). All the differences in the test AUCs between the LCRP model with GP-nodules and the LCRP model with real baseline nodules were not found to be significant for the three subgroups.

### Net Reclassification Improvement in Lung Cancer Risk Stratification

Table 3 shows the net reclassification index (NRI) when the LCRP model was deployed to the virtual GP-nodules to reclassify the lung cancer risks for subjects who were initially stratified by the Lung-RADs, Brock model, and LCRP model based on the real nodules from the baseline (T0) or the first follow-up LDCT scans (T1).

For 53 subjects with biopsy-proven lung cancer that were initially stratified into three risk groups by Lung-RADS with baseline nodules, the LCRP model using GP-nodules escalated 14 subjects (26.4%) to a higher risk (10 low-risk escalated to 5 medium- and 5 high-risk, and 4 medium- to high-risk), and 6 were de-escalated to lower risk, achieving an event NRI = 0.15 (*p* = 0.08 by Z-statistic). Among 397 subjects negative for lung cancer, 151 subjects (38.0%) initially categorized into groups of medium or high risk were de-escalated to a lower risk by the LCRP model using GP-nodules, and 57 subjects (14.4%) were escalated to higher risk, achieving a nonevent NRI of 0.24 (*p* < 0.001 by Z-statistic). The overall NRI of 0.38 indicated that the LCRP model using GP-nodules significantly improved the risk stratification (*p* < 0.001) when compared with the Lung-RADS at the time of baseline screening. Similar improvements by the LCRP model using GP-nodules were achieved when compared to the Brock model (NRI = 0.20, *p* = 0.03) and compared to the LCRP model using real baseline nodules (NRI = 0.20, *p* = 0.004). The difference in risk stratification compared to the LCRP model using real follow-up nodules did not reach statistical significance (NRI = 0.04, *p* = 0.62).

## 4. Discussion

Studying the evolution of lung nodules over time is essential for assessing the risk of lung cancer for patients with lung nodules detected on CT scans. Accurate and reliable risk stratification plays a pivotal role in the management of lung cancer that enables tailored approaches to screening, diagnosis, and treatment for personalized care (43). In this study, we developed a deep-learning-based generative GP-WGAN model that used lung nodule images in baseline LDCT scans to predict their growth or stability in 1-year follow-up LDCT scans. The results showed that the predicted 1-year follow-up nodule images by the GP-WGAN model achieved comparable performance with the corresponding real nodule images from 1-year follow-up scans for lung cancer diagnosis. Using GP-nodules to replace the baseline nodules, the LCRP model achieved a comparable test AUC of 0.827 with those using the real nodules either from the T0 baseline (AUC = 0.805, *p* = 0.099) or the T1 follow-up (AUC = 0.862, *p* = 0.071) LDCT scans, and significantly outperformed the Brock model (AUC = 0.754, *p* = 0.043) for early diagnosis of baseline nodules. The net benefit analysis (Table 3) showed that the reclassification of the baseline nodules by the LCRP model using GP-nodules consistently and significantly improved the risk stratification compared to the classification by Lung-RADS (NRI = 0.38, *p* < 0.001), the Brock model (NRI = 0.20, *p* = 0.03) and the LCRP model (NRI = 0.20, *p* = 0.004) using the real baseline nodules.

The findings demonstrated the feasibility of our proposed method in predicting the growth patterns of nodules with generated nodule images in follow-up exams. The GP-nodules enhanced the capability of a lung cancer risk prediction model in risk stratification at the baseline year. When compared with the LCRP model, Brock model, and Lung-RADS using real baseline nodules, the LCRP model with GP-nodules had a net gain in the identification of high-risk cases that could potentially impact mortality through early interventions. Furthermore, the GP-nodules improved the performance of the LCRP model in the classification of indeterminate nodules that were solid, spiculated, or within the 6 to 14 mm diameter range, as illustrated in Table 2. As these nodules would likely undergo follow-up exams in current clinical practice, better risk stratification is pivotal for reducing the indeterminate decision. Another advantage of generating the GP-nodules by our proposed method is it takes one step toward an explainable AI approach that may increase the radiologist’s understanding of the risk prediction by the LCRP model and enable them to make more appropriate diagnostic decisions after taking into consideration the AI recommendation and their own judgment. Therefore, our proposed approach has the potential to reduce unnecessary follow-up exams and improve early diagnosis of lung cancer at baseline screening. This is expected to have clinical significance as it may not only reduce patient anxiety and the associated healthcare costs but also aid the physician in making more informed decisions about the necessary level of aggressiveness in subsequent management steps. Moreover, the ability to generate GP-nodules may enable other computerized methods, for example, nodule segmentation and temporal analysis, to be sequentially used for specific tasks.

Designing an effective loss function is essential for developing image-generative models due to the inherent difficulties in determining image similarity, especially when predicting a future image in a stochastic process. We designed a generative loss function ($L_{G}$) for the GP-WGAN model by combining four loss functions. In the combined loss functions, the L1 loss is a widely used measurement in machine learning to minimize error by capturing low-frequency patterns that are relatively uniform with small changes. L1 has been demonstrated to improve image quality in various generative models for image generation and image-to-image translation [42,43]. Given the intrinsic correlation of objects in the image, a major limitation of L1 loss is its assumption of pixel independence in an image. Relying solely on this metric without considering structural information may not be able to reveal specific aspects of nodule growth patterns over time. A recent study [42] showed that using structural similarity index (SSIM) as a loss function during network training can preserve contrast in high-frequency regions compared to the L1 loss function. Other studies further demonstrated that the combination of L1 with SSIM outperformed standalone SSIM [42,44]. Besides combining L1 and SSIM in our generative loss function, we also incorporated a learned perceptual loss ($L_{LP})$ that utilized a pre-trained Resnet-18 model. The $L_{LP}$ could provide a more accurate representation of human perception, serving as a valuable metric for assessing dissimilarities in a high-level feature space that encompasses disparities in both content and style between images. The results showed that the GP-WGAN model with the newly designed loss function could generate more realistic nodule images through assessing the image quality of the generated nodules at the pixel level, structural level, and high semantic feature level.

Artificial intelligence, particularly in the form of generative models, has garnered significant interest and activity in recent years. However, owing to the lack of real physical models within the general framework, there is a growing concern about the reliability and ethical implications of “black-box” models, as well as the integrity of the data they generate. Challenges include ensuring data reliability and transparency, and developing methods to mitigate potential bias and discrimination. A considerable amount of work must be undertaken before these systems are ready for clinical applications, which includes conducting extensive external validation and prospective studies.

There are limitations in this study. Despite including all malignant nodule cases from the NLST dataset, which is relatively large compared to most studies in medical imaging [18], the subset that satisfied the requirements of the current study is still small when compared with the big data set used in conventional GAN-related research in the computer vision field. This may limit the learning of nodule progression patterns over time. For example, malignant nodules in our data set exhibited a wide variety of volume doubling times ranging from 6 months to several years. The relatively small training sample size of malignant nodules with a wide range of progression paces might result in a trained GP-WGAN model having a limited capability in predicting the nodule growth rates. Some examples are included in Appendix E. Another limitation is due to the lack of effective metrics to quantify the similarity between the virtual and real images. In this study, we adopted a task-based assessment approach by using a previously developed LCRP model as the downstream task to evaluate how the GP-nodules can improve the prediction of lung cancer risk. Further work is underway to conduct external validation, reader studies with radiologists, and prospective studies to assess the performance of the proposed model.

## 5. Conclusions

In conclusion, we developed a GP-WGAN model to predict lung nodule images in 1-year follow-up LDCT scans based on the baseline LDCT scans and studied its potential to improve the early diagnosis of lung cancer at the baseline screening year. The results demonstrated that the follow-up nodule images predicted by GP-WGAN could achieve comparable performance with the real nodules in follow-up LDCT scans, which indicates the use of the GP-WGAN model provides the opportunity to accelerate the clinical management of malignant nodules towards an earlier diagnosis of lung cancer, rather than waiting for the next screening CT.

## Figures and Tables

**Figure 1 cancers-16-02229-f001:**
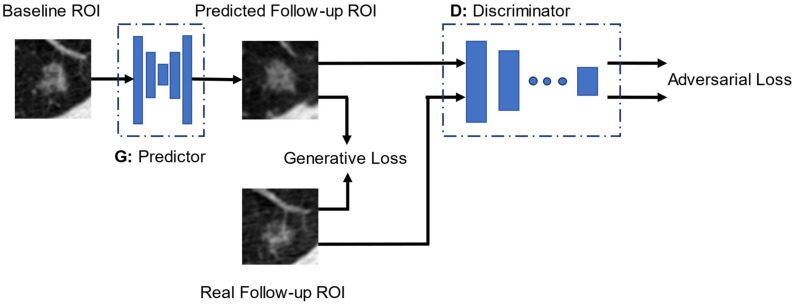
The adversarial framework for training the GP-WGAN model.

**Figure 2 cancers-16-02229-f002:**
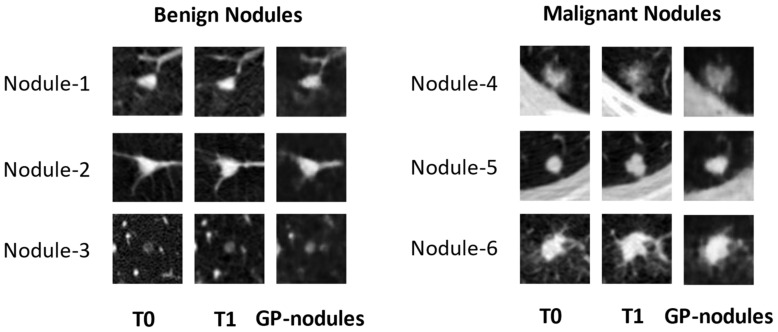
Examples of three benign (**left**) and three malignant (**right**) lung nodules from 6 subjects on baseline (T0) and 1-year follow-up (T1) LDCT scans. Each row presents an example nodule. For each nodule, the 1st column shows the ROI image of the nodule at T0, 2nd column shows its follow-up image at T1, and the 3rd column shows the GP-nodules generated from the T0 nodule shown in the 1st column by the GP-WGAN model. (**Left**): the benign nodules in T0, T1, and their predicted GP-nodules showed stability in size, attenuation, and smooth margins. (**Right**): the malignant nodules showed a trend of enlarged sizes in T1 and GP-nodules.

**Figure 3 cancers-16-02229-f003:**
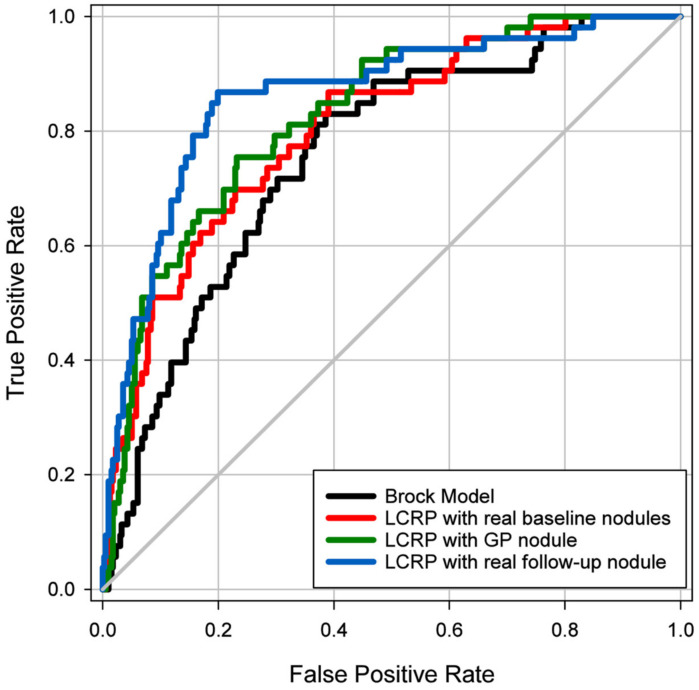
Test ROC curves of malignancy risk prediction by the LCRP model for classification of nodules generated by GP-WGAN (green line, AUC = 0.827 ± 0.028), real nodules from baseline (red line, AUC = 0.805 ± 0.031) and from 1-year follow-up LDCT scans (light-sky blue line, AUC = 0.862 ± 0.028). The Brock model was used to classify baseline nodules (black line, AUC = 0.754 ± 0.035) for comparison.

**Table 1 cancers-16-02229-t001:** NLST documented demographic characteristics, disease staging, and other radiologic factors.

Characteristic		Dataset (n = 1226)		Training/Validation Set (n = 776)		Test Set (n = 450)	
		Positive (n = 218)	Negative (n = 1008)	Positive (n = 165)	Negative (n = 611)	Positive (n = 53)	Negative (n = 397)
Age (y), mean ± SD		63.50 ± 5.09	61.89 ± 5.15	63.54 ± 4.97	61.55 ± 5.14	63.38 ± 5.45	62.40 ± 5.11
Gender	Female	94 (43.12%)	419 (41.57%)	71 (43.03%)	248 (40.59%)	23 (43.40%)	171 (43.07%)
	Male	124 (56.88%)	589 (58.43%)	94 (56.97%)	363 (59.41%)	30 (56.60%)	226 (56.93%)
Race	White	205 (94.04%)	939 (93.7%)	157 (95.15%)	566 (92.64%)	48 (90.57%)	373 (93.95%)
	Other	13 (5.96%)	69 (6.85%)	8 (4.85%)	45 (7.36%)	5 (9.43%)	24 (6.05%)
Ethnicity	Hispanic/Latino	0 (0.00%)	14 (1.39%)	0 (0.00%)	9 (1.47%)	0 (0.00%)	5 (1.26%)
	Other	218 (100%)	994 (98.61%)	165 (100%)	602 (98.53%)	53 (100%)	392 (98.74%)
Smoking	Current	120 (55.05%)	494 (49.01%)	92 (55.76%)	297 (48.61%)	28 (52.83%)	197 (49.62%)
	Former	98 (44.95%)	514 (50.99%)	73 (44.24%)	314 (51.39%)	25 (47.17%)	200 (50.38%)
Smoked, mean ± SD	Packs/yr.	65.10 ± 26.11	56.16 ± 25.06	65.07 ± 26.01	56.13 ± 25.37	65.17 ± 26.44	56.21 ± 24.58
	Avg/day	30.00 ± 11.93	28.48 ± 1201	30.05 ± 11.89	28.49 ± 11.63	39.85 ± 12.06	28.47 ± 12.47
	Years	43.86 ± 7.00	39.93 ± 7.23	43.83 ± 6.89	39.79 ± 7.23	43.96 ± 7.31	40.14 ± 7.23
Family cancer History, N (%)	Positive	50 (22.94%)	227 (22.52%)	39 (23.64%)	133 (21.77%)	11 (20.75%)	94 (23.68%)
Medical history	COPD	16 (7.34%)	46 (4.56%)	10 (6.06%)	28 (4.58%)	6 (11.32%)	18 (4.53%)
	Emphysema	28 (12.84%)	79 (7.74%)	19 (11.52%)	45 (7.36%)	9 (16.98%)	34 (8.56%)
TNM stage	Stage IA	115 (52.7%)		81 (49.1%)		34 (64.1%)	
	Stage IB	26 (11.9%)		19 (11.5%)		7 (13.2%)	
	Stage IIA	13 (6.0%)		11 (6.7%)		2 (3.8%)	
	Stage IIB	9 (4.1%)		9 (5.5%)		0 (0.0%)	
	Stage IIIA	20 (9.2%)		17 (10.3%)		3 (5.7%)	
	Stage IIIB	5 (2.3%)		4 (2.4%)		1 (1.9%)	
	Stage IV	24 (11.0%)		19 (11.5%)		5 (9.4%)	
	Other *	6 (2.8%)		5 (3.0%)		1 (1.9%)	
Histopathological subtype **	Bronchioloalveolar carcinoma	40 (18.3%)		30 (18.2%)		10 (18.9%)	
	Adenocarcinoma	92 (42.2%)		63 (38.2%)		29 (54.7%)	
	Squamous cell carcinoma	42 (19.3%)		35 (21.2%)		7 (13.2%)	
	Large cell carcinoma	4 (1.8%)		4 (2.4%)		0 (0.0%)	
	Non-small cell, other	21 (9.6%)		17 (10.3%)		4 (7.5%)	
	Small cell carcinoma	16 (7.3%)		14 (8.5%)		2 (3.8%)	
	Carcinoid	2 (0.9%)		1 (0.6%)		1 (1.9%)	
Margins	Spiculated	60 (27.5%)	80 (8.0%)	44 (26.6%)	42 (6.9%)	16 (30.2%)	38 (9.6%)
	Smooth	76 (34.9%)	720 (71.4%)	60 (36.4%)	444 (72.7%)	16 (30.2%)	276 (69.5%)
	Other ^†^	82 (37.6%)	208 (20.6%)	61 (37.0%)	125 (20.4%)	21 (39.6%)	83 (20.9%)
Internal characteristics	Soft Tissue	148 (67.9%)	773 (76.7%)	112 (67.9%)	475 (77.7%)	36 (67.9%)	298 (75.1%)
	Ground glass	42 (19.3%)	130 (12.9%)	33 (20.0%)	74 (12.1%)	9 (17.0%)	56 (14.1%)
	Mixed	19 (8.7%)	55 (5.4%)	14 (8.5%)	23 (3.8%)	5 (9.4%)	22 (5.5%)
	Other ^††^	9 (4.1%)	50 (5.0%)	6 (3.6%)	39 (6.4%)	3 (5.7%)	21 (5.3%)

Data are reported as number of participants (percentage) unless otherwise specified. NLST = National Lung Screening Trial. * Other conditions include occult carcinoma or cannot be assessed, as decided by NLST radiologists. ** Two/one subjects in the training/test set did not have this value. ^†^ Others include Poorly defined and Unable to determine, decided by NLST radiologists. ^††^ Others include Fluid/water, Fat, Other, and Unable to determine, decided by NLST radiologists.

**Table 2 cancers-16-02229-t002:** Test results achieved by the LCRP model and Brock model using virtual GP-nodules or real nodules for the classification of solid nodules, spiculated nodules, and nodules with sizes ranging from 6 to 14 mm in the independent test set.

Model\Criteria			All Nodule	Solid Nodule	Spiculated Nodule	Diameter 6 to 14 mm
n (Benign + Malignant)			397 + 53	298 + 36	38 + 16	267 + 38
LCRP + GP-nodule	AUC		0.827 ± 0.028	0.828 ± 0.037	0.850 ± 0.055	0.782 ± 0.041
	95% CI		(0.772, 0.883)	(0.762, 0.895)	(0.743, 0.958)	(0.702, 0.862)
LCRP + real follow-up	AUC		0.862 ± 0.028	0.864 ± 0.034	0.922 ± 0.037	0.826 ± 0.039
	95% CI		(0.806, 0.917)	(0.797, 0.931)	(0.848, 0.995)	(0.750, 0.902)
	*p*-value	LCRP + GP-nodule	0.071	0.091	0.150	0.077
		LCRP + real baseline	0.024	0.020	0.018	0.018
		Brock model	<0.001	0.002	<0.001	0.006
LCRP + real baseline	AUC		0.805 ± 0.031	0.793 ± 0.039	0.793 ± 0.067	0.749 ± 0.045
	95% CI		(0.744, 0.866)	(0.716, 0.870)	(0.661, 0.924)	(0.660, 0.838)
	*p*-value	LCRP + GP-nodule	0.099	0.058	0.105	0.083
		Brock model	0.146	0.249	0.007	0.201
Brock model + real baseline	AUC		0.754 ± 0.035	0.768 ± 0.040	0.595 ± 0.080	0.704 ± 0.042
	95% CI		(0.686, 0.823)	(0.690, 0.846)	(0.439, 0.751)	(0.621, 0.787)
	*p*-value	LCRP + GP-nodule	0.043	0.045	<0.001	0.048

**Table 3 cancers-16-02229-t003:** Net reclassification improvement (NRI) assessing the advantages of reclassifying subjects by the utilization of GP-nodules. The GP-nodules were predicted from their baseline LDCT scans using our GP-WGAN model. The 450 subjects in the test set were initially categorized by the Lung-RADs and Brock model in the groups of low, medium, and high risk, based on the nodules observed at the baseline scans. Similar assessments were also conducted for the LCRP model that used the real baseline and follow-up nodules. The net reclassification improvement for event NRI and nonevent NRI were calculated separately, and the overall NRI was the sum of these two values.

		Risk groups (Number of subjects, %)	Reclassified by LCRP model + GP-nodules			
Stratify real nodules			Low	Medium	High	
		Cancer (N = 53)	(N = 3, 5.66%)	(N = 21, 39.62%)	(N = 29, 54.72%)	
		Negative (N = 397)	(N = 164, 41.31%)	(N = 189, 47.61%)	(N = 44, 11.08%)	
Lung-RADS at baseline	Cancer (N = 53)	Low (N = 12, 22.64%)	2	**5 ^+^**	**5 ^+^**	event *NRI* = 0.15 *p* = 0.08
		Medium (N = 15, 28.30%)	0 ^−^	11	**4 ^+^**	
		High (N = 26, 49.06%)	1 ^−^	5 ^−^	20	
*NRI* = 0.38, *p* < 0.001	Negative (N = 397)	Low (N = 144, 36.27%)	89	46 ^+^	9 ^+^	nonevent *NRI* = 0.24 *p* < 0.001
		Medium (N = 136, 34.26%)	**67 ^−^**	67	2 ^+^	
		High (N = 117, 29.47%)	**8 ^−^**	**76 ^−^**	33	
Brock at baseline	Cancer (N = 53)	low (N = 5, 9.43%)	1	**4 ^+^**	**0 ^+^**	event *NRI* = 0.19 *p* = 0.02
		Medium (N = 27, 50.95%)	2 ^−^	14	**11 ^+^**	
		High (N = 21, 39.62%)	0 ^−^	3 ^−^	18	
*NRI* = 0.20, *p* = 0.03	Negative (N = 397)	Low (N = 162, 40.81%)	105	57 ^+^	0 ^+^	nonevent *NRI* = 0.01 *p* = 0.75
		Medium (N = 183, 46.10%)	**54 ^−^**	110	19 ^+^	
		High (N = 52, 13.10%)	**5 ^−^**	**22 ^−^**	25	
LCRP at baseline	Cancer (N = 53)	Low (N = 5, 9.43%)	3	**2 ^+^**	**0**	event *NRI* = 0.23 *p* < 0.001
		Medium (N = 29, 54.72%)	0 ^−^	19	**10 ^+^**	
		High (N = 19, 35.85%)	0 ^−^	0 ^−^	19	
*NRI* = 0.20, *p* = 0.004	Negative (N = 397)	Low (N = 158, 39.80%)	136	22 ^+^	0 ^+^	nonevent *NRI* = −0.03 *p* = 0.16
		Medium (N = 213, 53.65%)	**28 ^−^**	165	20 ^+^	
		High (N = 26, 6.55%)	**0 ^−^**	**2 ^−^**	24	
LCRP at 1-year follow-up	Cancer (N = 53)	Low (N = 3, 5.66%)	2	**1 ^+^**	**0 ^+^**	event *NRI* = 0.04 *p* = 0.60
		Medium (N = 23, 43.40%)	1 ^−^	14	**8 ^+^**	
		High (N = 27, 50.94%)	0 ^−^	6 ^−^	21	
*NRI* = 0.04, *p* = 0.62	Negative (N = 397)	Low (N = 155, 39.05%)	123	32 ^+^	0 ^+^	nonevent *NRI* = −0.003 *p* = 0.91
		Medium (N = 208, 52.39%)	**41 ^−^**	149	18 ^+^	
		High (N = 34, 8.56%)	**0 ^−^**	**8 ^−^**	26	

+ indicates an escalation, − indicates a de-escalation, and the bold font indicates a correct reclassification (either escalation or de-escalation). NRI = event NRI + nonevent NRI. The statistical significance was tested by Z-statistic (*p*-value) separately for NRI, event NRI, and nonevent NRI.

## Data Availability

In this study, we used the NLST dataset (https://cdas.cancer.gov/nlst/, accessed on 18 December 2015), which is publicly accessible, but obtaining permission from the NLST research team is required.

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
