# Peer review of "Enhancing Early Lung Cancer Diagnosis: Predicting Lung Nodule Progression in Follow-Up Low-Dose CT Scan with Deep Generative Model"

_cancers, 2024, doi:10.3390/cancers16122229_

Round 1

Reviewer 1 Report

Comments and Suggestions for Authors

This is an interesting study. The authors reported that the GP-nodules predicted by GP-WGAN model can forecast nodule appearance in the follow-up imaging from baseline LDCT scan, which enhances risk assessment during initial screening. However, the reviewer has the following concerns.

(1) What is the size range of lung nodules? The size of a lung nodule is an important factor in determining its risk of being cancerous.

(2) Many solitary pulmonary nodules (SPNs) show no significant change after only one year of follow-up, but may progress to malignancy after two years. Could the limited one-year follow-up period in this study influence the results?

(3) In Figure 2, what do the first, second, and third rows represent, respectively?

(4) What is the parameters of LDCT scanning. More information about the LDCT scans is described in Appendix A, but I can not find the Appendix A.

(5) For Stage IV patients, follow-up CT scans are performed one year later. However, it is unclear whether the patients received any relevant treatment during this year and whether the treatment would affect the results of this study.

Comments on the Quality of English Language

The paper is well organized, and the writing is professional and formal, which is easy to understand and the ideas are presented in a logical order, adhering to the rules of English grammar.

Author Response

Thank you very much for taking the time to review this manuscript. Please see the attachment for point-by-point response 

Reviewer 2 Report

Comments and Suggestions for Authors

This research paper uses a deep generative model to predict lung nodule progression in follow-up LDCT scans, with the goal of  enhancing the early lung cancer detection accuracy. However, there are still some areas that need to be improved.

1. The title should be more concise and clearer. One possible suggestion is : "Enhancing Early Lung Cancer Detection: Predicting  Follow-up Nodule Development with Deep Generative Models".

2. At the end of the introduction section, it is recommended that a summary of innovations and contributions of the research conducted be presented.

3.Please provide specific information about the CT equipment and acquisition parameters.

4.The notation "mm3" should have the "3" as an exponent..

5.Nodules are 3D, though images of these in this research are in 2D. Selecting an image patch has effects on results. What rules did the author have in selecting 2D patches?

6.  Please elaborate on the network architecture parameters of GP-WGAN.

7.The reliability of the generated images needs to be evaluated. The authors can provide a similarity assessment between the generated and the follow-up images, and determine whether experts can distinguish between the generated images and real images.

8. The ethical considerations of using AI in medical diagnostics should be discussed.

Author Response

Thank you very much for taking the time to review this manuscript. Please see the attachment for point-by-point response.

Round 2

Reviewer 2 Report

Comments and Suggestions for Authors

The author has addressed the raised issues and recommends the publication of the work.